# A Novel Bio-Inspired Ag/3D-TiO_2_/Si SERS Substrate with Ordered Moth-like Structure

**DOI:** 10.3390/nano12183127

**Published:** 2022-09-09

**Authors:** Jingguo Yang, Florian Ion Tiberiu Petrescu, Ying Li, Dandan Song, Gang Shi

**Affiliations:** 1Key Laboratory of Synthetic and Biotechnology Colloids, Ministry of Education, School of Chemical and Material Engineering, Jiangnan University, Wuxi 214122, China; 2Department of Mechanisms and Robots Theory, Bucharest Polytechnic University, 060042 Bucharest, Romania

**Keywords:** SERS, bio-inspired, ultra-sensitivity, recyclability

## Abstract

This paper reports a novel method to fabricate a bio-inspired SERS substrate with low reflectivity, ultra-sensitivity, excellent uniformity, and recyclability. First, double layers of polystyrene spheres with different particle sizes were assembled on the surface of a silicon wafer to act as a moth-like template. Second, through the template sacrifice method, the TiO_2_ film with a three-dimensional moth-like eye structure was induced by the double-layer polystyrene spheres in the previous step, and its microscopic morphology showed a high degree of order. Finally, Ag nanoparticles were assembled on the TiO_2_ film to form a bio-inspired SERS substrate. This ordered bio-inspired structure can not only reduce reflection, but also reinforce the uniformity of hotspot density, which helps to improve the sensitivity and uniformity of the Raman signal. This bio-inspired SERS substrate can detect R6G molecules at a concentration as low as 1.0 × 10^−10^ mol/L, and its enhancement factor (EF) can reach 6.56 × 10^6^. In addition, the composite of Ag and TiO_2_ can realize the photocatalytic degradation of R6G and then realize the recyclability of the SERS substrate.

## 1. Introduction

Surface-enhanced Raman scattering (SERS) is an important method for the analysis of trace substances, which has been widely applied to the detection of chemical substances, biomolecules, and environmental pollutants [1,2,3,4,5]. It has the advantages of excellent specificity and high sensitivity to the target analytes [6,7,8]. Generally, the Raman signal enhancement is usually attributed to the electromagnetic field effect [9,10,11] of the substrate, and the noble metals on it play a leading role [12,13,14]. The surface microstructure of the SERS substrate has a great impact on the Raman signal intensity, and the three-dimensional (3D) structure usually has the stronger Raman signal [15,16,17]. On the one hand, the 3D structure can provide more effective hotspots [18,19]. On the other hand, the 3D structure with a high specific surface area can adsorb more target analytes, thus improving the Raman signal. Furthermore, the 3D structure has excellent antireflective performance, which can increase the absorption efficiency of excited light, and then improve the Raman signal [20,21].

Most SERS substrates can only be used once because the analytes adsorbed on the surface of the SERS substrate are not easily removed. Therefore, current research usually focuses on the development of recyclable SERS substrates [22,23,24]. TiO_2_ is an excellent photocatalytic material, of which composites with noble metals (such as Au, Ag, Cu, etc.) not only improve the photocatalytic efficiency, but also expand the SERS capability. This noble metal/TiO_2_ composite can achieve the recyclability of the SERS substrate [25,26,27]. In addition to recyclability, the uniformity of the Raman signal is one of the important evaluation factors for SERS substrates. Fabricating an ordered micro/nano-structure is an effective way to achieve the uniformity of the Raman signal. The ordered micro/nano-structure (such as spheres, rods, bowls, etc.) of TiO_2_ and its composite substrates can be obtained by the sol–gel method, hydrothermal method, and chemical vapor deposition [28,29,30]. Among them, the metal film-covered ordered TiO_2_ sphere (MOTS) SERS substrate has always been focused on due to its stability, uniformity, and recyclability. However, the traditional MOTS SERS substrate still needs improvement to enhance the detection sensitivity.

The antireflective ability of moth-eye structures can effectively improve the absorption efficiency of incident light, so it is worth considering how to apply this to the MOTS substrate to enhance the Raman signal intensity. Here, a two-step colloidal spheres template method is proposed to fabricate a Ag/3D-TiO_2_/Si composite SERS substrate with an ordered moth-like eye structure. This SERS substrate can not only improve the intensity and uniformity of Raman signal at the same time, but also realize the recyclability of the substrate. The SERS substrate contains noble metal Au or Ag, which have high costs. In the detection process, organic molecules adsorbed on the traditional SERS substrate are difficult to remove, which will have a non-negligible impact on the subsequent detection. Therefore, it can only be used once, which limits the promotion and application to families, such as the detection of drinking water, fruits, vegetables, milk, meat, and so on. Based on the consideration of the above problems, this study developed a reusable SERS substrate to reduce its use cost, which provided the possibility of applying the SERS base to home detection.

## 2. Materials and Methods

### 2.1. Chemicals and Materials

Acetone (CH_3_COCH_3_), chloroform (CHCl_3_), ethanol (CH_3_CH_2_OH), hydrochloric acid (HCl), n-butyl titanate (C_16_H_36_O_4_Ti), stannous chloride dihydrate (SnCl_2_·2H_2_O), sodium dodecyl sulfate (SDS), rhodamine 6G (R6G), and potassium persulfate (KPS) were purchased from Sinopharm Chemical Reagent Co., LTD., Shanghai, China. Silver nitrate (AgNO_3_) and styrene (C_8_H_8_) were purchased from Sigma Aldrich Trading Co., LTD., St. Louis, MO, USA. All reagents were used directly without further purification. The Si wafers (p-type (100)) were obtained from Youyan Guigu, Beijing, China. Distilled water was used.

### 2.2. Fabrication of 2D and 3D Polystyrene (PS) Templates

PS spheres with different particle sizes (1570 nm and 180 nm) were synthesized by the emulsion polymerization method [31,32,33]. Then, the 2D and 3D PS templates were fabricated by the self-assembly method, and the detailed procedures are as follows.

Firstly, 10 wt% 1570 nm of PS spheres suspension was diluted with the same amount of ethanol and then dropped onto the water surface. Secondly, a drop of 5% SDS was added to form a closely packed monolayer of PS spheres, and then it was assembled on the Si wafer. Thirdly, the solvent was evaporated to obtain the 2D PS template. After heating at 90 °C for 5 min, the adhesion between PS microspheres and the substrate was increased. Fourthly, 180 nm of PS spheres were continued to assemble on the above 2D PS template in accordance with the above method. Finally, the 3D PS template was received after drying under natural conditions.

### 2.3. Fabrication of 2D-TiO_2_/Si and 3D-TiO_2_/Si

TiO_2_ sol was obtained by classical hydrolysis of n-butyl titanate and diluted to 0.213 mol/L [34]. Then, 30 uL of TiO_2_ sol was spin-coated onto the 2D or 3D colloidal template. The above samples were placed in a muffle furnace at 450 °C for 2 h with a heating rate of 1 °C/min. When TiO_2_ was coated on the 2D PS template and calcined, the obtained sample was named 2D-TiO_2_/Si. When TiO_2_ was coated on the 3D PS template and calcined, the obtained sample was named 3D-TiO_2_/Si.

### 2.4. Fabrication of Ag/2D-TiO_2_/Si and Ag/3D-TiO_2_/Si

After 0.1 mol/L SnCl_2_ solution was mixed with 0.02 mol/L AgNO_3_ solution, 2D-TiO_2_/Si or 3D-TiO_2_/Si were immediately immersed in the mixed solutions and reacted at 60 °C for 8 h. Thus, a certain amount of Ag NPs was deposited on the TiO_2_ surface [35]. The SERS substrates after Ag deposition on 2D-TiO_2_/Si and 3D-TiO_2_/Si were named Ag/2D-TiO_2_/Si and Ag/3D-TiO_2_/Si, respectively. From the experimental preparation to the successful fabrication of Ag/3D-TiO_2_/Si, the whole process took 29 h. The sample area can be up to 4 cm × 4 cm.

### 2.5. Raman Measurements

The substrates were immersed in R6G ethanol solution for 30 min and then washed with deionized water. The Raman spectrum was measured immediately after the above substrates were dried with nitrogen.

### 2.6. Photocatalytic Measurements

The photocatalytic capability of the substrates (1.0 cm × 2.0 cm) was measured by the degradation of R6G under irradiation of the 300 W Xe light source (200 nm < λ < 2500 nm). The substrates were immersed in 2 mL of the R6G solution (10^−4^ mol/L, neutral condition) and stirred in the dark for 30 min to ensure adsorption equilibrium before the illumination. After the illumination, the R6G solution was immediately measured using a UV–vis spectrophotometer.

### 2.7. Photoelectrochemical (PEC) Measurements

The photocurrent intensity and the electrochemical impedance evaluations were measured in a standard three-electrode system [36,37]. PEC measurements were performed with the substrates, Pt filament, and Ag/AgCl as the working electrode, the counter electrode, and the reference electrode, respectively. The PEC capability was measured by the electrochemical analyzer with 0.5 mol/L Na_2_SO_4_ solution as the electrolyte, and the simulated solar light source (100 mW/cm^2^) was irradiated in the 1.0 × 1.0 cm^2^ area from the front of the working electrode.

### 2.8. Characterizations

SEM was performed using a Hitachi S4800 (Hitachi, Tokyo, Japan) microscope. UV–vis spectra were recorded using a UV–vis spectrometer (UV-3600 plus, Shimadzu, Kyoto, Japan). The photocurrent response and the resistance effect were recorded by an electrochemical station (CHI-660, Shanghai Chenhua, Shanghai, China). Photocatalytic degradation was carried out under a 300 W xenon lamp (CEL-HXF 300, CEAuLight Co., Ltd., Beijing, China) as the visible light source. The Raman spectra were recorded at room temperature on a Raman system (Renishaw inVia, Renishaw plc, Gloucestershire, UK) equipped with a charge-coupled device (CCD) detector and a 532 nm laser of 0.05 mW. The spot diameter of the laser was 2 μm. Signal accumulation with an integration time of 10 s was used to collect the spectra for all Raman measurements.

## 3. Results and Discussion

### 3.1. Fabrication and Surface Characterization of SERS Substrate

The fabrication process of the Ag/3D-TiO_2_/Si substrate with a 3D moth-like eye structure is shown in Figure 1. First, monolayers of PS spheres with diameters of 1570 nm and 180 nm were assembled on the Si wafer successively by the interfacial assembly and transfer technique. Figure 1a,b are the SEM images of 1570 nm monolayer PS spheres, which are uniform in size, compact in arrangement, and highly ordered in a large area. Second, 180 nm PS spheres were assembled on the surface of 1570 nm PS spheres through a similar method described above. Figure 1c,d show that each 1570 nm PS sphere is covered with 30–50 of 180 nm PS spheres. This two-size hierarchical structure imitates the moth-eye structure, which was used as a template to fabricate the bio-inspired SERS substrate. Third, the TiO_2_ sol was coated on the surface of the above template, and the PS spheres were then removed by high-temperature calcination to obtain the TiO_2_ film with an ordered moth-like eye structure, as shown in Figure 1e,f. The calcined TiO_2_ is in anatase form, as shown in Appendix A. Finally, Ag NPs were deposited on TiO_2_ by the in situ reduction method of Sn^2+^, and the obtained sample is referred to as Ag/3D-TiO_2_/Si. Ag NPs with a particle size of 20–60 nm are densely packed around the porous TiO_2_, as shown in Figure 1g,h. Through the EDS characterization of Ag/3D-TiO_2_/Si (Figure 2), it is further proved that the bio-inspired SERS substrate has been fabricated successfully.

### 3.2. Raman Performance of Ag/3D-TiO_2_/Si

It can be seen from Figure 3a that the Raman signal of R6G adsorbed on the Ag/3D-TiO_2_/Si substrate is significantly stronger than that of others, which is mainly attributed to the following two aspects. First, the 3D structure of Ag/3D-TiO_2_/Si increases the number of hotspots compared with the 2D structure of Ag/2D-TiO_2_/Si. According to the SEM images of Ag/3D-TiO_2_/Si (Figure 1g,h) and Ag/2D-TiO_2_/Si (Appendix A), the former substrate has a larger number of regular porous structures than the latter. As a result, the specific surface area is larger, and the loaded Ag NPs are relatively more abundant; thus, Ag/3D-TiO_2_/Si will create more hotspots and more adsorption sites, which is conducive to the enhancement of the Raman signal. Second, the 3D moth-like eye structure of Ag/3D-TiO_2_/Si conforms to the equivalent medium theory [36,38,39,40,41], which will make the refractive index change gradually from the air to the substrate, and then the incident light reflection will be reduced; in other words, the incident light absorption will be enhanced. Therefore, the reflectivity of Ag/3D-TiO_2_/Si is significantly lower than that of Ag/2D-TiO_2_/Si, as shown in Figure 3b, which is conducive to the enhancement of the Raman signal. It is worth noting that Ag/3D-TiO_2_/Si has a wide wavelength absorption band at 400–600 nm, which is attributed to the plasmon resonance absorption [42,43,44] on the substrate surface, which is beneficial to the enhancement of the Raman signal [45].

The detection sensitivity is also an important property in evaluating the performance of the SERS substrate. Here, the Ag/3D-TiO_2_/Si substrates were immersed in different concentrations of R6G ethanol solutions, and their corresponding Raman spectra are shown in Figure 4a. With the decrease in R6G concentration, the Raman signals are gradually decreased. The concentration of R6G detected by the SERS substrate is as low as 10^−10^ mol/L. This indicates that the Ag/3D-TiO_2_/Si substrate has high sensitivity. In addition, the uniformity of the SERS substrate is also an important property. Figure 4b shows the Raman spectra of R6G (10^−4^ mol/L) at 20 random points of the Ag/3D-TiO_2_/Si substrate. The relative standard deviation (RSD) of the Raman characteristic peak at 1368 cm^−1^ is 9.0%, implying that Ag/3D-TiO_2_/Si has excellent uniformity. This is because the 3D structure of the Ag/3D-TiO_2_/Si substrate is highly ordered, and the Ag NPs are evenly distributed.

The enhancement factor (*EF*) is an important parameter to evaluate the Raman enhancement of the SERS substrates, and the specific formula is as follows [33,46]:(1)$$EF=\frac{I_{\mathrm{SERS}}{\times N}_{0}}{N_{\mathrm{SERS}}{\times I}_{0}}$$
where *I*_SERS_ and *I*_0_ are the Raman peak (1368 cm^−1^) intensities of R6G on the Ag/3D-TiO_2_/Si substrate and the Si wafer, respectively. *N*_SERS_ and *N*_0_ are the number of R6G molecules on the Ag/3D-TiO_2_/Si substrate and the silicon wafer, respectively.

The R6G molecules on the substrate are considered to be distributed uniformly; thus, *N*_SERS_ and *N*_0_ can be estimated by the following formula:(2)$$N=\frac{N_{A}{MVS}_{\mathrm{laser}}}{S_{\mathrm{sub}}}$$
where *N*_A_ is Avogadro’s constant, *M* is the concentration of the R6G solution, *V* is the volume of the droplet, *S*_sub_ is the substrate area (1 cm^2^), and *S*_laser_ is the laser point area (1 μm^2^). Here, 10 μL of 10^−8^ mol/L R6G solution is dropped onto the Ag/3D-TiO_2_/Si substrate, while for the Si wafer, 10 μL of 10^−2^ mol/L R6G solution is dropped onto it. Based on the peak intensity of 1368 cm^−1^ in the Raman spectra of Figure 5 and combined with Equations (1) and (2), *EF* = 6.56 × 10^6^ is evaluated.

### 3.3. Photocatalysis and Recyclability of Ag/3D-TiO_2_/Si

According to the dark state adsorption test, the concentration of R6G is almost unchanged after adsorption–desorption equilibrium. Figure 6a shows the absorption spectra of R6G in the process of photocatalytic degradation by the Ag/3D-TiO_2_/Si substrate under simulated solar radiation. With the extension of time, the R6G concentration decreases gradually until it no longer changes after two hours. It can be found from Figure 6b that the Ag/3D-TiO_2_/Si substrate has the fastest catalytic degradation rate and the best photocatalytic capability on R6G. The excellent photocatalytic degradation of R6G by Ag/3D-TiO_2_/Si can ensure the self-cleaning of the SERS substrate so as to realize recyclable SERS detection. The Raman signal of the Ag/3D-TiO_2_/Si substrate was tested after the adsorption of R6G molecules with a concentration of 10^−4^ mol/L (detection step). After the substrate was treated with the simulated solar irradiation for 2 h, the Raman signal almost completely disappeared (self-cleaning step). After four cycles of detection/self-cleaning, the Raman signal intensity of the substrate remained almost unchanged, corresponding to RSD (the Raman characteristic peak at 1368 cm^−1^) of 4.1%, as shown in Figure 6c. These results fully prove the excellent recyclability of the ordered 3D SERS substrate.

In order to further analyze the reasons for the excellent photocatalytic performance of the Ag/3D-TiO_2_/Si substrate, linear sweep voltammograms (LSVs) and impedance characterization were performed, as shown in Figure 7a,b. In the dark, the photocurrent densities of Ag/3D-TiO_2_/Si, Ag/2D-TiO_2_/Si, and 3D-TiO_2_/Si are almost zero, and their corresponding curves are very similar. By comparing the Ag/3D-TiO_2_/Si substrate and the 3D-TiO_2_/Si substrate, the photocurrent density of the former is higher than that of the latter, and the internal resistance of the former is lower than that of the latter. This is because Ag NPs inhibit the photogenerated electron–hole pairs recombination of TiO_2_, which is conducive to the improvement in photocatalytic efficiency. In addition, by comparing the Ag/3D-TiO_2_/Si substrate and the Ag/2D-TiO_2_/Si substrate, the photocurrent density of the former is higher than that of the latter, and the internal resistance of the former is lower than that of the latter. This is because the 3D moth-like eye structure has excellent antireflection performance, which is beneficial to improving the absorption efficiency of the incident light, thus increasing the number of TiO_2_ photogenerated electron–hole pairs, and then the photocatalytic efficiency of the SERS substrate will be improved.

## 4. Conclusions

In summary, a 3D TiO_2_ film with moth-like eye structure was fabricated on the Si wafer by the colloidal self-assembly method and then covered with a layer of uniformly distributed Ag NPs to form a novel SERS substrate. This bio-inspired structure not only reduced the reflection, but also provided more hotspots in the laser radiation area. R6G as low as 1.0 × 10^−10^ mol/L was still detected by the Ag/3D-TiO_2_/Si substrate, and the corresponding EF reached 6.56 × 10^6^. The ordered structure of Ag/3D-TiO_2_/Si resulted in the uniformity of Raman signals (RSD = 9.0%). Meanwhile, the Ag/3D-TiO_2_/Si substrate was recyclable due to its excellent photocatalysis. This study developed a reusable SERS substrate to reduce its use cost, which provides the possibility of applying SERS base to home detection, such as the detection of drinking water, fruits, vegetables, milk, meat, and so on.

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
