# Peer review of "A Novel Bio-Inspired Ag/3D-TiO2/Si SERS Substrate with Ordered Moth-like Structure"

_nanomaterials, 2022, doi:10.3390/nano12183127_

Round 1

Reviewer 1 Report

Publish as is.

Author Response

Thank you very much!

Reviewer 2 Report

1. The guiding significance to the practical application should be provided at the end of the Introduction and Conclusions, in order to make the idea more convincing.

2. The resolution of Scheme 1 should be improved.

3. I am confused about Figure 2b and Si element mapping in Figure 2c, they should not be so similar, there is some mistakes here, and all the element mapping images in Figure 3c are so fuzzy, cannot tell anything, new images should be provided.

4. Eq (1) in Line 195 and Eq (2) in Line 202, why all the equations looks so fuzzy? Please re-write them.

5. There are 6 legends in Figure 7 (a), but only 4 curves are plotted. Please double check and re-plot it.

6.  I would like to suggest going through the manuscript more carefully for clarity, syntax and correctness.

Author Response

Thanks for your questions and suggestions. Please see the attachment.

Reviewer 3 Report

In this paper, a novel bio-inspired Ag/3D-TiO2/Si SERS substrate with ordered moth-like structure was developed. After the following problems have been solved, the work is proposed to be published in Nanomaterials.

1. The distribution of O, Ti and Ag element mapping in Fig. 2 is not clear, is it because the content of elements is too small? If not, change the element color to make it easier for readers to read.

2. Formulas (1) and (2) are ambiguous.

3. What is the total time period of the process and what is the largest substrate size that can be fabricated within the time of reference? Add this in discussion to help readers know more information of the fabrication process.

4. Recyclability is a critical factor for SERS substrate. Please give the relative standard deviation to illustrate the recyclability intuitively.

5. The English language needs to be improved.

Author Response

(The authors gave the same response as above.)

Round 2

Reviewer 2 Report

The authors only made a perfunctory effort on the guiding significance to the practical application, they added the same paragraph "This reusable SERS substrate can reduce the cost of Raman detection and is expected to be promoted in the field of food safety." at the end of the Introduction and Conclusions, which is far from adequate.

Author Response

Thanks for your suggestion. Please see the attachment.

Round 3

Reviewer 2 Report

Accept